# Growth Regulation in the Larvae of the Lepidopteran *Pieris brassicae*: A Field Study

**DOI:** 10.3390/insects14020167

**Published:** 2023-02-09

**Authors:** Sebastian Baraldi, Emanuele Rigato, Giuseppe Fusco

**Affiliations:** Department of Biology, University of Padova, Via U. Bassi 58/B, 35131 Padova, Italy

**Keywords:** canalization, compensatory growth, developmental stability, fluctuating asymmetry, geometric morphometrics, phenotypic variation, post-embryonic development

## Abstract

**Simple Summary:**

Size and shape are important features of most living beings, and the capacity of the organism to regulate size and shape during growth, containing the effects of developmental disturbances of different origins, is considered a key adaptation. Through a morphological study on the larvae of the cabbage butterfly *Pieris brassicae*, reared in conditions close to the natural environment, we found that the regulatory mechanisms for limiting the effects of developmental disturbances, known to be effective under strictly controlled, very stable laboratory conditions, are also effective during growth under more natural environmental conditions. This study may contribute to better characterization of these regulatory mechanisms and their combined effects in the developmental interactions between the organism and its environment.

**Abstract:**

Size and shape are important determinants of fitness in most living beings. Accordingly, the capacity of the organism to regulate size and shape during growth, containing the effects of developmental disturbances of different origin, is considered a key feature of the developmental system. In a recent study, through a geometric morphometric analysis on a laboratory-reared sample of the lepidopteran *Pieris brassicae*, we found evidence of regulatory mechanisms able to restrain size and shape variation, including bilateral fluctuating asymmetry, during larval development. However, the efficacy of the regulatory mechanism under greater environmental variation remains to be explored. Here, based on a field-reared sample of the same species, by adopting identical measurements of size and shape variation, we found that the regulatory mechanisms for containing the effects of developmental disturbances during larval growth in *P. brassicae* are also effective under more natural environmental conditions. This study may contribute to better characterization of the mechanisms of developmental stability and canalization and their combined effects in the developmental interactions between the organism and its environment.

## 1. Introduction

Size and shape are two important life history traits of most organisms [1]. Accordingly, the capacity of the organism to regulate size and shape during growth, by buffering against the effects of different sources of developmental disturbances, is considered a key feature of the developmental system and has been recorded in many taxa [2].

A key concept in the study of size and shape regulation during growth is that of *target ontogenetic trajectory*, which can be defined as the series of character states, through all the developmental stages of an individual with a specific genotype, in a specific environment, in the absence of any disturbance [3]. This derives by extension from the concept of ‘target phenotype’ [2,4], where the ontogenetic trajectory embodies a more inclusive notion of the phenotype [5]. During development, growth trajectories, both in size and shape, tend to diverge from the target trajectory, under the influence of external factors such as temperature or nutrition, or as a consequence of stochastic effects on the developmental processes [6].

Regulatory mechanisms able to limit the effects of different perturbing factors are generally distinguished based on their effect of phenotypic variation. *Canalization* mechanisms reduce environmental effects *on* the target ontogenetic trajectory, i.e., its variation with environmental factors, while *developmental stability* mechanisms buffer against developmental noise, i.e., against variation *around* the target phenotype, reducing phenotypic dispersion for any combination of environmental conditions [4]. The two forms of regulation can also work in concert [7].

The possibility of detecting operating regulative mechanisms based on observational data is grounded on the expectations of theoretical models of growth (reviewed in [8]). A steady increase of phenotypic variance, both in size and shape, across ontogeny at the population level is expected simply because of the cumulative effect of random variation in individual growth rates at each stage. In addition, random deviations within individuals from the expected body symmetry, known as fluctuating asymmetry (FA) [9] are also expected to increase across ontogeny. In organisms with bilateral symmetry, random deviations from left-right symmetry, or bilateral FA [10], are employed to investigate developmental stability, i.e., the ability of an organism to buffer random perturbations of its developmental process [4,11]. In arthropods, since growth of the exoskeleton largely occurs stepwise, paced by the moult cycle, the regulatory mechanism can take the form of *compensatory growth* (also called *targeted growth*), wherein individuals adjust their own growth trajectory stage-by-stage, keeping it close to the target ontogenetic trajectory [3].

In a recent rearing experiment conducted in our laboratory, by adopting a longitudinal study-design (i.e., by producing a dataset consisting of the measurements of the same individual in subsequent stages; [12]) and through morphometric analyses based on geometric morphometrics, we found evidence of size and shape regulation during the larval development of the cabbage butterfly *Pieris brassicae* [8]. In that study, larvae were reared individually at constant temperature (25 °C), humidity (50% RH) and photoperiod (16L:8D), and fed with a semi-synthetic diet. The study was designed to mainly detect compensation of developmental noise in very stable environmental conditions. However, the efficacy of the regulatory mechanism under greater environmental variation remains to be explored.

Here, through rearing larvae of the same species in the field, and by adopting identical measurements of size and shape variation, we show that the regulatory mechanisms for containing the effects of developmental disturbances during larval growth are also effective under more natural and variable environmental conditions.

*Pieris brassicae* (Linnaeus, 1758), the cabbage butterfly (or the large white), is distributed throughout Europe, Asia and North Africa, where it is considered a major pest on cruciferous vegetables. Detailed information is available on morphology, growth and phenology of the immature stages (e.g., [13,14,15]), as well as on the influence of several external factors on larval development (e.g., [16,17]). More recently, this information has been supplemented by our ontogenetic studies on size, allometry and timing during post-embryonic development [18], and the patterns of growth regulation [8]. *P. brassicae* has basically an invariable number of larval stages (five). This is important for studies such as the present one, where confidence in the assignment of specimens to homologous developmental stages (i.e., where each larval stage corresponds to a homologous segment of ontogeny for all specimens) is the basis of ontogenetic analysis [3].

The capacity of developmental systems to contain variation due to internal or external causes has been reported for many taxa. However, despite recent advances in understanding the molecular and physiological mechanisms of growth regulation in a few insect model species, observational data on size and shape regulation are still relatively scarce, especially under natural conditions of growth. This study, by combining the results of laboratory and field observations, may contribute to better characterization of the mechanisms of developmental stability and canalization during post-embryonic development.

## 2. Materials and Methods

The study is based on a dataset of morphometric measurements of the head capsule on 300 field-reared specimens of the lepidopteran *Pieris brassicae* during the five larval stages (L1–L5) (3 specimens by 5 larval stages, by 20 egg clutches). Size and shape development were quantified and analysed through geometric morphometric methods [19]. The study has a non-longitudinal design, i.e., each stage is represented by a different set of individuals [12]. This is an unavoidable constrain of the quasi-natural rearing conditions, where, because of their gregarious habit, the larvae were allowed to grow in groups. Hereafter, for simplicity, we will label with ‘field’ the methods and results of the present study, whereas those of [8,18] will be labelled as ‘laboratory’.

### 2.1. Rearing and Collection of Larvae

Larvae were reared starting from 20 egg clutches laid by as many females from a stock population at the insect farm Smart Bugs (Ponzano Veneto, Italy). The stock derives from natural populations of Northern East Italy and includes more than 5000 reproductive individuals. This is sustained by a constant supply of individuals from natural populations (at least one per generation), providing the appropriate level of gene flow to ensures low levels of drift and inbreeding [20]. In parallel, 20 cabbage plants were grown in a nearby plot, without using any pesticides or herbicides. Clutches and plants were paired randomly.

Rearing was carried out in quasi-natural conditions, only preventing the confounding effects of parasitoid infection on growth and sample depletion caused by ant harvesting. The day after deposition, egg clutches were placed individually in Petri dishes (Ø 35 mm), laid on a cabbage leaf fragment from the paired plant. Petri dishes were placed in a rack system located outside the Insect Farm. After moulting to the second stage for the majority of the larvae from a given clutch, larvae were moved to larger transparent containers with a vented lid (plastic box 23 × 13 × 8 cm). Animals were checked every two days, for hatching first and for moulting thereafter. At the same time, leaf fragment and larval droppings were removed, and new freshly picked leaves from the paired plant were provided. Upon hatching or moulting to the next stage of the majority of the larvae from a given clutch, three specimens were picked up randomly and individually fixed in a test tube filled with 70% ethanol. Most larvae moulted to the fifth stage within 10–11 days of hatching.

Meteorological data for the rearing period (June 2021) were recorded by a meteorological station (run by the Veneto Regional Agency for Environmental Prevention and Protection) about 5 km away from the experiment site (Appendix A). Both temperature and humidity were within the normal range for the place and the season. Daily mean temperature varied between 20.5 and 25.3 °C, with a daily excursion in the range of 8.4–15.8 °C, while relative humidity varied daily between about 40 and 100%.

### 2.2. Landmark Choice and Data Acquisition

Measurements were taken on the cephalic capsule of larval stages L1 to L5, specifically on the two genae plus the frons (hereafter, *frons*). These were analysed through geometric morphometrics [21]. To make data directly comparable with previous studies, the set of landmarks chosen was identical to the configuration adopted to build the laboratory dataset (Figure 1).

The 19 landmarks are approximately coplanar, limiting the error deriving from the projection of their three-dimensional arrangement onto the plane of the image [22]. One medial landmark was placed at the vertex of the frons, whereas the other 18 were positioned at the basis of 9 pairs of idionymic setae (i.e., setae homologous across stages and individuals within the species).

Larvae heads were photographed with a digital camera (LEICA DFC 420) mounted on a stereoscope (LEICA MZ12.5), using a zoom objective with declining magnification from 6.3× to 1.25×, from stage L1 to L5. Images were acquired through the Leica Application Suite software (ver. 2.8.1) at a size of 2588 × 1943 pixels. To control for measurement error [9], two images were taken separately for each specimen, following independent placements of the animal under the stereomicroscope, and landmarks were digitized twice on each of the two images by the same person (SB), in two independent working sessions, using TPSDig 2 (ver. 2.31; [23]). Each specimen was thus represented by four different sets of landmark coordinates. The program tpsUTIL (ver. 1.81; [23]) was used to build the final (.NTS) data file by combining all data into a single dataset.

### 2.3. Morphometric Analyses

The landmark configuration of the frons presents a type of bilateral symmetry where the symmetry axis is found within the structure itself (*object symmetry*), and morphometric analyses were carried out in conformity with this feature [9,24].

Landmark raw coordinates were rigidly transformed and scaled through a generalized Procrustes superimposition, producing a measure of linear size and scale-independent shape coordinates [25]. Size was estimated as the centroid size (*CS*), the square root of the sum of squared distances of all landmarks from their centroid [26]. Procrustes analysis separates symmetric and asymmetric components of shape variation [27] (Appendix A).

Statistical analyses were performed in MorphoJ (ver. 1.07a; [28]) and Statgraphics Centurion (ver. 19.2.02), with auxiliary calculations in Microsoft Excel 365.

#### 2.3.1. Size Analyses

The combined effect of clutch and plant on larval size at each stage was checked with one-way ANOVAs.

The logarithm of per-moult growth rate at each of the first four stages (*lnGR_1_–lnGR_4_*) was calculated as the difference of the natural logarithms of the *CS* (*lnCS*, averaged by individual and stage) between consecutive stages (this is equal to the logarithm of the postmoult/premoult *CS* ratio). The average per-moult growth rate was calculated as the exponential of the arithmetic mean of the four *lnGR*s (this is equal to the geometric mean of the four *GR*s) [29].

Departure from growth at a constant rate (Dyar’s rule; [30]) was tested with an ANOVA on the lnGR at the four moults, and the magnitude of the deviation was measured with the index of conformity to Dyar’s rule (*IDC*; [29]). This ranges from 0 (maximal monotonic departure from Dyar’s rule, total growth increment realized in only one stage) to 1 (perfectly constant growth rate).

The Levene’s test for equality of variances was used for probing the ontogenetic progression of size variance.

#### 2.3.2. Shape Analyses

Analyses of shape variation were based on the Procrustes ANOVA [24,31], a parametric two-factor general linear model. Total shape variation is partitioned into the main effect of ‘individual’ (i.e., variation among individuals, the symmetric component), ‘side’ (i.e., directional asymmetry, non-random variation between the two sides), the interaction ‘individual-by-side’ (i.e., random variation between the two sides, FA) and measurement error (residual).

The combined effect of clutch and plant on larval shape at each stage was checked by including an extra factor (‘clutch/plant’) in the Procrustes ANOVAs.

In the ontogeny of a given species, *ontogenetic allometry* denotes size-related shape changes across different developmental stages, whereas *static allometry* indicates size-related shape differences within the same developmental stage. Both relationships were studied through multivariate regression analysis [32], regressing symmetric shape on the logarithm of centroid size (*lnCS*). A permutation test (10,000 rounds) was used to estimate the significance of the relationships.

The ontogenetic progression of the magnitude of symmetric shape variation was assessed with the Hartley’s test for equal variance and the Cochran’s C test for variance outliers. Both tests are applicable with equal sample sizes (as in our case) and when data are distributed approximately normally, a realistic assumption for population-level shape data [9].

Fluctuating asymmetry at the level of the population was quantified with the *FA10* index. For each stage, this is computed as the square root of the difference between the mean squares of the ‘individual-by-side’ term and the mean squares of ‘measurement error’ term [33]. Since *FA10* is a variance estimate, variation of FA across ontogeny was assessed with the Hartley’s test for equal variance.

## 3. Results

### 3.1. Size and Shape Change across Ontogeny

Size and shape ANOVAs at each stage showed that the combined factor ‘clutch/plant’ had a significant effect on both size and shape ontogenetic progression (non-significant only for shape in L1; Appendix A). Sampling of larvae from different clutches and rearing on different plants was aimed at reproducing variation in natural conditions (see Section 4), but our study was not designed for specifically assessing the effects of these factors. All subsequent analyses were thus carried out by pooling all the individuals of each stage.

The average per-moult grow rate was 1.62 (Figure 2A), with growth rates (*GR*s) at each stage decreasing progressively and significantly from 1.71 in L1 to 1.54 in L4 (Figure 2B, ANOVA on *lnGR*, F = 25.82, *p* < 0.0001). Sizable differences in *GR* translate into a relatively low value of the index of conformity to Dyar’s rule, (*IDC* = 0.95), reflecting an appreciable deviation of size progression at a constant rate.

Multivariate regressions of the symmetric component of shape on size (*lnCS*) across all stages (ontogenetic allometry) was significant (*p* < 0.0001) (Figure 3).

Size accounted for 29.80% of the total amount of shape variation. Ontogenetic shape variation mainly consists of a peripheral elongation of the structures in the upper part of the genae, associated with a medial elongation of the lower part of the frons in the strict sense. Contrary to ontogenetic allometry, static allometry was in general modest (explaining 2–6% of size variation within each stage), but for stage L1, where size accounts for 13.20% of shape variation (*p* < 0.0001). 

Overall, the patterns of size and shape changes across stages (average sizes, growth rates and allometry) do not differ appreciably from those observed in laboratory rearing.

### 3.2. Size Regulation

Size variance showed a significant decrease across the five larval stages (Levene’s test, W = 4.21, *p* = 0.0025; Figure 4). 

Size variance at a given stage in the absence of compensation can be estimated as the sum of the variances of two independent random variables, namely size and growth rate at the previous stage [8]. Accordingly, the expected ontogenetic progression of size variance in the absence of compensation for our sample was calculated by setting the expected size variance in L1 equal to the observed value, and by iterating the addition of the average individual growth rate variance at each successive stage. This value (0.00058), which cannot be obtained from field (non-longitudinal) data, was derived from laboratory data. This estimate should be considered as very conservative, since it was recorded under highly controlled rearing conditions. The comparison of the observed size variance at stage L5 with the corresponding benchmark value calculated in the absence of size regulation reveals a level of compensation of 74%, which can nonetheless be considered an underestimation.

Size variance at each stage for the field dataset is slightly larger than for the laboratory, but significantly only at stages L1 and L2 (Fisher’s F tests, F = 3.04, *p* < 0.0001 and F = 1.74, *p* = 0.0154).

### 3.3. Symmetric Shape Regulation

Symmetric shape variation (factor ‘individual’) accounted for 68–75% of total variation (total sum of squares) at each stage and showed no increase across ontogeny (Figure 5). Shape variance in L1 is significantly larger than that in all successive stages (Cochran’s C test, C = 0.29, *p* < 0.0001), while it does not differ significantly across the latter (Hartley’s test, F_max_ = 1.04, *p* = 0.81).

At each stage, symmetric shape variance for the field dataset is significantly larger than for the laboratory (Fisher’s F tests, F = 1.91–1.28, *p* < 0.0002), and the difference is conspicuous at stage L1.

### 3.4. Fluctuating Asymmetry Regulation

Procrustes ANOVAs showed significant FA in all stages (‘individual-by-side’ interaction factor; (*p* < 0.0001). Fluctuating asymmetry at each stage explained 93–97% of total asymmetry variation (sum of squares of factor ‘side’ + ‘individual-by-side’), the remainder of which can be attributed to directional asymmetry.

*FA10* indexes showed no increase across stages (Figure 6). Shape variance in L1 is significantly larger than that in all successive stages (Cochran’s C test, C = 0.25, *p* = 0.003), while it does not differ significantly across the latter (Hartley’s test, Fmax = 1.28, *p* = 0.71).

*FA10* at each stage in the field is slightly larger than in the lab, but significantly only at stage L1 (Fisher’s F test, F = 1.35, *p* < 0.0001).

## 4. Discussion

In [8], through a detailed morphometric analysis of longitudinal growth data on a laboratory-reared sample of the lepidopteran *P. brassicae*, we found evidence of regulatory mechanisms able to restrain size and shape variation, including bilateral fluctuating asymmetry, across the first four larval stages. Here, by analysing a comparable morphological dataset obtained from field-reared specimens of the same species, extended to all five larval stages, we show that the regulatory mechanisms of *P. brassicae*, able to limit the effects of developmental disturbances, are also effective in more natural conditions of growth, with higher levels of environmental variation.

Per-stage growth rates and allometry patterns do not differ appreciably between laboratory and field samples. However, sizable differences emerge for developmental timing. The average duration of larval development up to the moult to the fifth stage in the field (10–11 days), for an average temperature of about 23 °C across the rearing period (Appendix A), is close to a value (10.2 days) interpolated from time measurements in [13] at 20 and 25 °C, but considerably shorter than in our laboratory study (15.4 days; [18]). The longer developmental times in the latter are explained by the combined effects of the synthetic diet [34] and the growth of the larvae in isolation from other individuals [16] (*P. brassicae* larvae are gregarious). The rearing conditions that we settled in the field, allowing the larvae to graze in a group on cabbage leaves, certainly recreated more natural conditions of growth. However, the soundness of the comparison between laboratory and field data is not impaired by these differences, since developmental timing seems not to be involved in regulation (see below), and per-moult growth patterns in both size and shape are the same in the two datasets.

### 4.1. Regulation of Size

The ontogenetic progression of size variance clearly shows the mark of compensatory growth. Unfortunately, without information on the individual growth trajectory of single specimens, as is the case in a non-longitudinal dataset like ours, the entity of the compensation cannot be quantified, and evidence of compensation rests on the observation of a non-significant increase in size variance across stages. However, by assuming an average individual variation in growth rates across stages no less than that observed in the laboratory, where rearing conditions were kept stable, we estimated a substantial reduction in size variance. This is in the order of 74% at the fifth larval stage, at the end of a process of seven-fold growth in linear body size and more than 300-fold growth in body mass from hatching.

Size regulation by compensatory growth has been documented for several animal taxa, including many insects (see [3,8], and references therein). This is an ancient property of animal developmental systems, at least for arthropods, as it has reported for some segments of ontogeny also in a few trilobite species [35,36]. However, besides a few model species (e.g., *Manduca*, among the lepidopterans; review in [37,38]), the underlying developmental mechanisms of growth regulation are in general poorly understood [39,40]. What is known for *P. brassicae* is that size compensation seems not to be accomplished by regulating stage duration, but rather by modulating per-time growth rates [8]. This stands out with respect to the high potential of time regulation, especially in holometabolous insects, where mass growth within stages is approximately exponential [41], and even small changes in stage duration can importantly affect body size, especially at later stages, with direct consequences on adult body size [42].

### 4.2. Regulation of Shape

Neither symmetric shape variance, nor fluctuating asymmetry has been shown to increase across larval stages. Contrary to size, for which a quantitative model for the expected ontogenetic increase in size variance is available [8,43], no comparable quantitative models are available for shape. Nonetheless, both symmetric and asymmetric shape variation are expected to increase across developmental stages, because there are multiple possible directions of variation in the shape space, and departure in a given direction at one stage neither compensates for nor precludes deviations in other directions at later stages. Therefore, unless variation is continually reduced, new variation is expected to add to that accumulated in earlier stages [44]. Based on this rationale, similarly to ontogenetic variation in size variance, a non-significant increase in shape variance can be interpreted as evidence of shape regulation.

Different theoretical models make discordant predictions about the ontogenetic variation in FA, entailing amplification, constancy or reduction (review in [8,45]). This is paralleled by discordant observational reports on different species and characters, which document an increment of FA across ontogeny [46,47], a constancy of FA [48,49,50], and even a decrease in FA [51,52]. The question is evidently in need of further investigations, as many variables, of different nature (e.g., taxon, character, developmental stage, environmental conditions) are possibly involved in the divergent behaviours of distinct developmental systems.

### 4.3. Concluding Remarks

Results of the analyses on the field dataset largely confirm the results obtained with the laboratory dataset. There is evidence that *P. brassicae* has the capacity to control both size and shape ontogenetic progression by limiting the effects of developmental disturbances also under semi-natural, but nonetheless variable, environmental conditions. As expected, levels of variation are slightly higher for the field dataset because differences in diet (leaves from distinct plants) and meteorological conditions (due to slight variations in developmental timing) add to putatively similar levels of genetic variation (similar number of clutches, 19 and 20, in the two studies). However, the direction of the response of the developmental system is the same in the two growth conditions.

The only major difference between the two datasets concerns the first larval stage, which shows significantly larger variation in the field for all size and shape variables. We further scrutinized this result with a series of additional measurements and tests, looking for possible confounding factors arising from sampling or from the design of our study. However, individuals at the extremes of L1 size and shape distributions are not the same, and do not come from a subset of ‘deviant’ clutches. We also checked for possible growth within stages (measuring a disjoined set of L1 larvae collected at 0, 12, 24, 36 and 48 h after hatching), putatively due to the modest hardening of the cuticle of the cephalic capsule in the delicate larvae of the first stage, but we found none. We thus conjecture that the larger amount of variation in the first larvae of the field dataset actually depends on the more variable rearing conditions experienced during the 3–4 days from egg deposition to hatching. This could be the result of a modest heterochrony in the time of hatching with respect to the progression of embryonic development. Minimal shifts in the embryonic/postembryonic divide [3] are expected to affect the size as well as the shape of the first larva, in consideration of the more conspicuous static allometry at the first stage compared to later stages.

Growth regulation, both in size and shape, is expected to differ to some extent between laboratory and natural conditions, due to the greater heterogeneity of environmental influences in the latter [50]. However, differences might affect growth in opposite directions. On one side, buffering mechanisms effective in the laboratory might result in less effective containment of natural environmental effects, resulting in ontogenetic increments of size and/or shape variation. Conversely, certain regulatory mechanisms, unexploited under laboratory conditions because they need stronger environmental stimuli to be triggered by, might be elicited under more effective solicitation, resulting in a higher level of compensation. The similarity between field and laboratory growth patterns in our study shows that the effectiveness of the buffering mechanism in the field is not lesser than that observed in the laboratory. Even without considering the prominent drop in variation between stages L1 and L2, cautiously considering the possibility that this might be due to some other undetected effects, compensation is apparent in all subsequent stages. Actually, for size variation, there is some indication that compensation is even more consistent and efficient in the field than in the laboratory stages.

Another issue about the differences between laboratory and field studies is that under laboratory conditions it is mainly developmental stability mechanisms, aimed at reducing developmental noise, that are expected to operate; meanwhile, under natural (or quasi-natural) conditions, these are expected to work together with canalization mechanisms [11]. In the few dedicated studies, there is an argument about whether the underlying processes of developmental stability and canalization are the same or different [2,53,54], something we cannot address through the design of our experiment. The amount of environmental variation experienced by the field-reared individuals of our sample rests on different nourishing plants and differences in weather conditions due to slight heterochronies in individual ontogenetic progression. Although this variation is possibly reduced with respect to real conditions in the wild, it is nonetheless significantly larger than that in the laboratory. Thus, although the laboratory response to developmental noise was possibly mainly concerned with developmental stability, a certain amount of control of pertinence of canalization is expected to have operated in the field.

In conclusion, the larval growth of *Pieris brassicae* shows the marks of developmental regulation, which has proven to be effective under a range of environmental conditions that is relevant for the species, as it is close to the natural conditions of growth. While providing information on the regulative performances of *P. brassicae’s* developmental system, the present work may contribute to better characterization of the species’ mechanisms of developmental stability and canalization, and their combined effects, in the developmental interaction between the organism and its environment.

## Figures and Tables

**Figure 1 insects-14-00167-f001:**
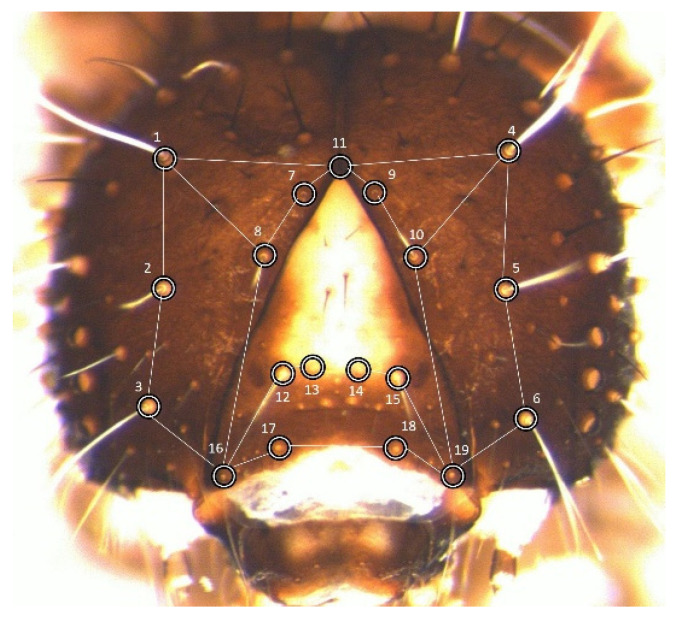
Position of landmarks (circles) on the head of *P. brassicae* larvae.

**Figure 2 insects-14-00167-f002:**
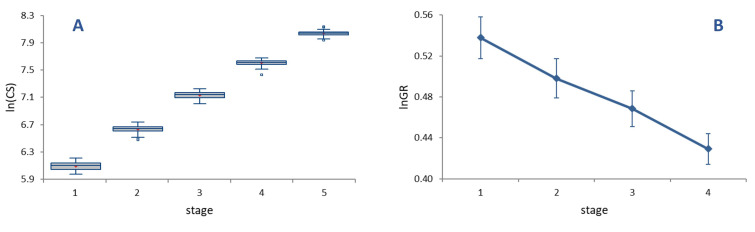
(**A**) Ontogenetic progression of size (*lnCS*) during the five larval stages in *P. brassicae*. Boxes represent the interquartile interval, with median (transverse line) and mean (red cross); vertical lines are ranges of variation, to the exclusion of outliers (dots). (**B**) Log-transformed per-stage growth rates (*lnGR*). Bars are 95% confidence intervals.

**Figure 3 insects-14-00167-f003:**
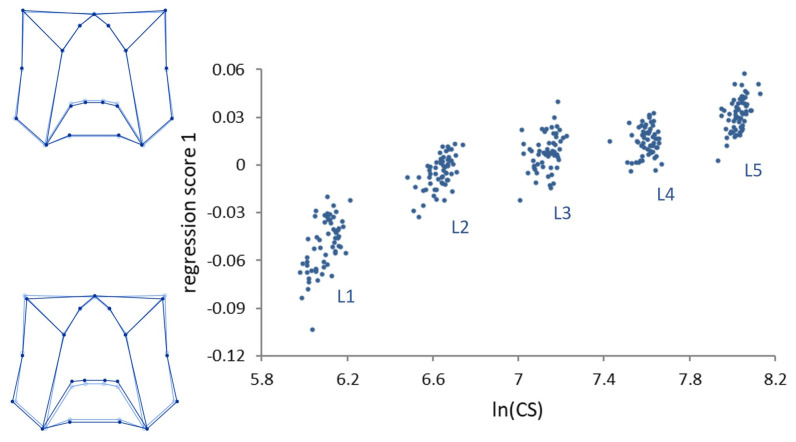
Multivariate regression of shape on size (*lnCS*) during the five larval stages in *P. brassicae*. Dark blue wireframes show the pattern of shape variation along the vertical axis with respect to the average shape (light blue wireframes).

**Figure 4 insects-14-00167-f004:**
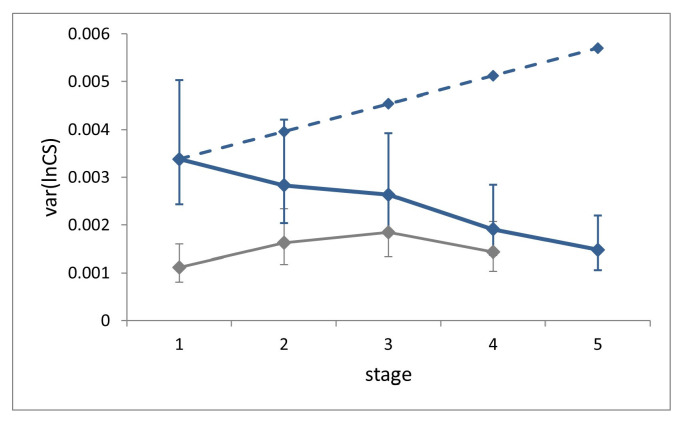
Ontogenetic progression of size variance (var(*lnCS*)) during the five larval stages in *P. brassicae*. Continuous blue line, field dataset; grey line, laboratory dataset (from [8]). The dashed line is the expected progression of size variance in the absence of compensation for the field dataset. Bars are 95% confidence intervals.

**Figure 5 insects-14-00167-f005:**
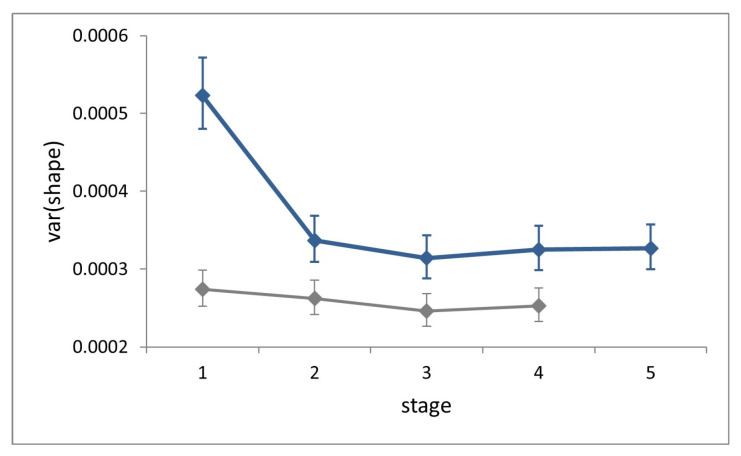
Ontogenetic progression of symmetric shape variance during the five larval stages in *P. brassicae*. Continuous blue line, field dataset; grey line, laboratory dataset (from [8]). Bars are 95% confidence intervals.

**Figure 6 insects-14-00167-f006:**
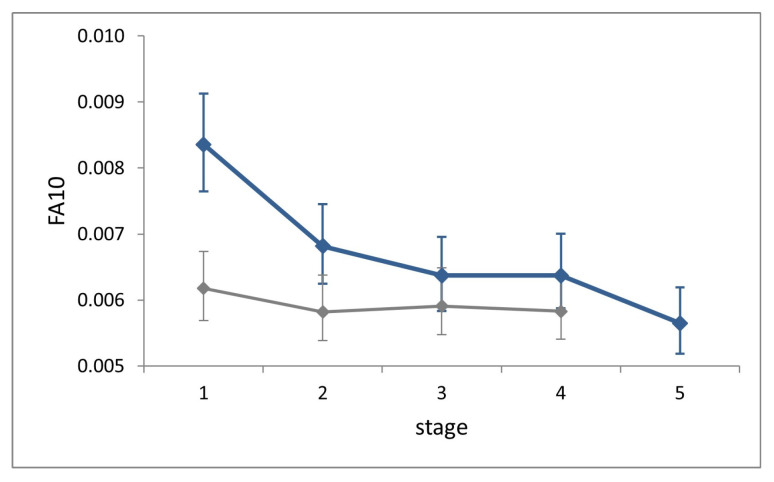
Ontogenetic progression of fluctuating asymmetry (measured as *FA10*) during the five larval stages in *P. brassicae*. Continuous blue line, field dataset; grey line, laboratory dataset (from [8]). Bars are 95% confidence intervals.

## Data Availability

All data are in the article and the Appendix A.

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
