# Peer review of "Growth Regulation in the Larvae of the Lepidopteran Pieris brassicae: A Field Study"

_insects, 2023, doi:10.3390/insects14020167_

Round 1

Reviewer 1 Report

This is traditional biology on insect development but interesting work. 

some concerns 

1) All figures should be improved and consistent in presentation, such as Figure 2, a,b should be merged insider of sub-figure.

2) Can you showed the rearing condition(temperature) in the lab and field in main file or supplementary. Temperature is key factor in development and growths for individuals.  Might it can be used for explaining the shape variation.

3) In this manuscript, you focus on the size and shape. You might connect those information with development rate. Might you can find some good ideas. 

4) In figure 4, why the bigger difference on size variation between lab and field ? please more discussion and how did you estimate the dash line ?

5) in context, families can be replaced by "group " or other words ?

6)  the bigger concern- you get the 20 egg clutches from many females, this different background of gene might influence the shape or size ? please give some explaining in method or discussion.

Author Response

Se attached file

Reviewer 2 Report

Review of the manuscript: "Growth regulation in the larvae of the lepidopteran Pieris brassicae".

The manuscript presents morphometric analysis of Pieris brassicae larvae. It is overall well-written, however I have a few doubts listed below:

1.      In Materials and methods there is a statement:  Heretofore, for simplicity, we will label with ‘field’ methods and results of the present study, whereas those of [8, 18] will be labelled as ‘laboratory’.” (lines 105-108). As I understand, it means that the Authors in reviewed manuscript are re-analysing the results they presented in their previous papers? If yes, why do it again in this paper? In my opinion, this is a kind of self-plagiarism. Moreover, the Authors refer to 2 of their previous papers in which the methods for laboratory research is described, and one of them [references 18] does not have the open access option and cannot be read freely.

2.      In my opinion there is no clearly formulated research hypothesis. In the manuscript there is a statement: “This study, by combining the results of laboratory and field observations, may contribute to a better characterization of the mechanisms of developmental stability and canalization during post-embryonic development” (lines 95-98). Which factors of both experiments could affect on morphometric parameters – temperature, diet? Maybe this aspect should be analysed in your research, especially that the title you proposed is: “Growth regulation  in the larvae …".

3.      The entire second paragraph in the Introduction (lines 38-46) is self-plagiarism (see references 8 and its Introduction).

Author Response

Se attached file

Round 2

Reviewer 2 Report

The manuscript "Growth regulation in the larvae of the lepidopteran Pieris brassicae" has been improved to warrant publication in Insects.